## [Transparent Peer Review file · Nature Communications]

Ecological Partitioning Enables Phage–Antibiotic Cooperation in a Human *Pseudomonas* Infection

Corresponding Author: Dr Dwayne Roach

Version 1:

Reviewer comments:

Reviewer #1

(Remarks to the Author)

The manuscript "Ecological and immune pressures shape outcomes of precision phage therapy in advanced cystic fibrosis lung disease" by Dr Roach and colleagues present a well-executed and clearly written case study detailing intravenous phage therapy using a two-phage cocktail in combination with ciprofloxacin to treat an elderly cystic fibrosis patient following colistin-associated acute kidney injury. This work is clinically significant and contributes valuable insights into the in vivo biology of phage therapy. Although case studies on phage treatment have been published previously, they continue to be highly relevant given the complexity of host-pathogen-phage interactions and the personalized nature of such interventions. The inclusion of virome analysis and the assessment of systemic neutralizing antibodies are particularly innovative aspects of this study. However, my enthusiasm is somewhat tempered by the lack of *P. aeruginosa* gene expression data, which would have helped elucidate how phage therapy affected bacterial fitness in vivo. Additionally, the systemic immune response could have been explored in greater depth. For example, while the authors propose that the response to phage PY02 may represent a secondary immune response, this interpretation remains speculative without supporting data. Analyzing the phage-specific IgM-to-IgG ratios over time could have provided a clearer picture of the immunological dynamics and enhanced the clinical relevance of the findings.

Major points to be addressed:

1. Statistics:

Obviously, with $n = 1$ not much statistics can be performed. However, when the authors did perform statistics, they forgot to mention the statistical test employed. Please correct that at least in Fig. 3,4 and S2. and check where else it applies.

2. Phylogeny and *Pseudomonas* biological fitness:

I am in full agreement with the authors, since phages fail to fully eradicate the pathogens in CF patients, even when used in combination with antibiotics, phage therapy driven alterations in bacterial biological and microenvironmental fitness become critically important to develop the optimal CF therapy.

In line 219-..., the authors conclude, based on Fig. 3b (phylogeny) that virulence and related traits of *Pseudomonas* strains may be altered. In addition, they evaluate and find growth limitations and hyperpigmentation (Fig. 3e, S8). However, bacterial gene expression analysis is lacking to experimentally confirm these assumptions. Do certain clades acquire specific genetic features that limited their virulence? Which bacterial genes are changing under phage pressure?

Alternatively, key elements of *Pseudomonas aeruginosa* virulence should be assessed, such as pyocyanin and elastase production (both of which contribute to tissue damage in cystic fibrosis). Biofilm formation and the ability to adhere to epithelial cells should also be considered, as these are prerequisites for persistent infection. Since both phages target LPS, potential changes in LPS structure or expression should be investigated. (Could LPS changes may explain the observed shift toward Pbnavirus phages increase after treatment?).

3. Systemic immunity:

Again, I fully agree with the authors that the development of neutralizing antibodies is a highly relevant finding and highlights the importance of immunomonitoring during phage therapy. However, the authors do not convincingly demonstrate that the immune response to P1A represents a primary response, while the response to PY02 constitutes a secondary response. Assessing the IgM-to-IgG ratios and timing of Ig levels over time could help clarify this distinction. Such an analysis would provide a more complete picture and determine whether the response to PY02 is truly secondary, or instead represents a primary response that is accelerated due to e.g. higher immunogenicity.

4. Virome:

Third, I agree with the authors that analyzing the virome after prolonged treatment is highly important. However, the observation remains largely descriptive. The authors should experimentally investigate what biological changes or

advantages in the lung environment enable the outgrowth of Pseudomonas phages.

Reviewer #2

(Remarks to the Author)

This paper presents a study on the use of phage therapy combined with antibiotics to treat a severe pulmonary exacerbation caused by MDR *Pseudomonas aeruginosa* infection in a 76-year-old patient with cystic fibrosis (CF). The combination therapy resulted in rapid improvement in lung function, reduced airway obstruction, and a 100-fold decrease in bacterial load. This study highlights that phage therapy efficacy might be rethought and that combined with antibiotics it offers promising potential for treating chronic multidrug-resistant infections in cystic fibrosis.

However, the data supporting this claim are limited in their generalization for a couple of reasons. The first major point is the choice of phages used in the study. The two phages P1A and PYO2 seem to be present in the patient before treatment which indicates that the phage-host relationship is already ongoing and it evolves in a balanced ecosystem where the phage is not effective against one of the strains (in this case the mucoid strain). Therefore, the likelihood of eradication is limited and the authors should clarify the reason behind the choice of those two phages and not a more complex cocktail of phages with established lysis activity against the strains of *P. aeruginosa* from this patient. The authors should also give a clear description of the different strains of *P. aeruginosa* present in the patient with the MLST and if possible the cgMLST of the strains to explain their clonality and differences.

The second major point is the usage of the metagenomic data. The authors limited their analysis to the relative and absolute quantification of the species. The absolute quantification of the species based on the human read counts is debatable, especially during an exacerbation. The amount of human cells in the sputum during an inflammation will increase tremendously based on neutrophil migration and increased of other inflammatory cells making the normalisation unreliable to the least. A spike-in method at the time of extraction would be more accurate. Furthermore, the amount of data collected would allow the author to do strain typing/monitoring to evaluate the abundance of the different clones during the treatment which would be highly valuable measures for the host-phage dynamic understanding.

In summary, while the case is interesting and the treatment had some clinical benefits, the choice of phages is questionable and should be explained. The fact that the phage is endogenous and the mucoid strain was already resistant to both phages should be highlighted as a limitation. Furthermore, the manuscript would benefit from a deeper analysis of the metagenomic data to get the most out of it, especially regarding the host-phage interaction.

I also have some minor comments :

Line 451 : ceftalozone-tazobactam should be ceftolozane-tazobactam
In Figures 3 B and C: the color-code of the isolate should be explained

Reviewer #3

(Remarks to the Author)

This is an interesting case study of a patient administered phage therapy to target *P. aeruginosa* in the CF lung. Many valuable conclusions came from the many analyses conducted by the authors. My concerns below mostly speak to accuracy when describing the unique aspects of the work, compared to other studies preceding it. Overall, I hope that the authors find the major and minor concerns as useful feedback.

Major concerns

Line 93. The staggered treatment strategy was designed to minimize antagonism between antimicrobials. But I believe this assumed that delivering the phage 2 days ahead of antibiotic would cause the virus to disappear, and the study showed this assumption was not supported. Can the authors revisit/discuss this assumption given the outcome of the study?

Line 118. After therapy was discontinued, WBC and platelet counts returned to 'near baseline'. Is this outcome consistent with the observation of adaptive immunity targeting the phage itself? Otherwise, I am confused how both can be true.

Lines 169-170. I am unfamiliar with the PCA analysis and its underlying assumptions. It seems to be a correlative analysis where the studied factors are assumed to drive the correlations, whereas it is not possible to eliminate the role of unknown factors. This is understandable given the limitations of testing the hypothesis in a treated human. If the authors can provide greater context for the assumptions and/or limitations of this analysis, it would benefit the reader's understanding.

Lines 249-262. The poor ability for PYO2 to replicate within the treated patient suggests that phage P1A dominated any efficacy attributable to phage delivery. Thus, it is unclear whether any claim of a 'cocktail' effect truly exists in this study, which is not made evident to the reader. Although the *in vitro* preliminary research argued that it would be a good idea to co-administer the two phages, it seems difficult to conclude that any different outcome would have occurred if just phage P1A were used on its own. The authors should discuss this outcome, as it would be a valuable opportunity to address how *in vitro* prelim work can or cannot be used to design *in-patient* therapy.

Lines 270-271. Related to the above point, the authors argue that neutralization of PYO2 occurred earlier in the patient than P1A. Was there no neutralization test of pre-existing antibodies conducted prior to administering the phages? Otherwise, it seems that this confirmation would have suggested a different treatment design, and removal of PYO2 from the cocktail based on evidence that it should be neutralized when delivered.

Lines 278 – 283. This first paragraph of the Discussion seems too strongly worded. I do not think the results 'reframe what is possible and necessary' in phage therapy of pwCF, given the many studies that preceded it. To my understanding, the main novelty is the virome analysis, though I did not scour the literature to confirm. Some studies have observed phage neutralization, whereas others have not. This suggests that the choice of phage used in treatment can affect whether neutralization can occur, which in turn should affect whether establishment in the virome is likely. By this logic, the choice of the two phages (i.e., the particulars of phage biology) in the current study was clearly important, but I caution the authors against claiming that we have learned anything very general in this respect. Better to not over claim here, as we learn a lot from each personalized phage therapy case in biomedicine (including here), and should be careful about trying to generalize. Basically, I am arguing that the authors are contradicting their own statements in Lines 350-351, where they warn against concluding generality from this study.

Lines 285-292. This paragraph may need updating, in terms of more holistically framing the study in context of similar ones that are now in the literature.

Lines 302-304. The observation of phage ability to steer traits in target bacteria seems evident in the current study, but I worry that not citing other prior work constitutes over-claiming. There is mounting evidence for this outcome in the published literature including use of phage therapy when targeting *P. aeruginosa* pulmonary infections in pwCF, for example.

Line 310. Indeed, this is a nice cautionary tale that we should understand how phage therapy does (and does not) work when targeting infections that include subpopulations of mucoid phenotypes. But I did not see any evidence in the current study that the current phage cocktail was any better at reducing biofilms. Thus, I am wondering why the authors are forwarding cocktails as a needed solution here, as opposed to simply emphasizing that targeting biofilms is a good idea, regardless of the number of administered viruses.

Line 359. Again, I am unclear why the authors are claiming that their study shows success in high-risk patients. Hasn't this been true in many prior studies as well? I do not think they can claim priority here, given the thousands of other successful cases to date, especially the subset of elderly patients without other treatment options that were treated with phage therapy.

Minor concerns

Line 37. Why is the CF lung referred to here as a 'hostile' environment? Without context, this adjective seems anthropomorphic and unnecessary.

Line 60. The Chan et al. 2025 Nature Medicine article and any other recent relevant studies on phage therapy targeting *P. aeruginosa* pulmonary infections in pwCF can be cited here.

Version 2:

Reviewer comments:

Reviewer #1

(Remarks to the Author)

I would like to congratulate the authors to the excellent study. All my points have been addressed.

To adhere to Nat Comms 'Sex and Gender Equity in Research – SAGER – guidelines' the authors should include the sex of the patient in either title or abstract.

Reviewer #2

(Remarks to the Author)

The authors answered all my comments from my previous revision accurately. When corrections was not possible they acknowledge the limitations and gave their rationale. From my side the manuscript is now proper for publication.

Reviewer #3

(Remarks to the Author)

The authors nicely addressed all of my concerns and I have no further comments.

Reviewer Response

Phage–Antibiotic Cooperation Emerges Through Ecological Partitioning in Heterogeneous *Pseudomonas* Infection

Previously titled: Ecological and immune pressures shape outcomes of precision phage therapy in advanced cystic fibrosis lung disease

Authors: Tiffany Luong, Lukeman Kharrat, Kevin Champagne-Jorgensen, Jennifer A. Melendez, David Pride, Douglas J. Conrad, and Dwayne R. Roach

We thank the reviewers for their constructive feedback, which has substantially strengthened and expanded the manuscript. In response, we performed new analyses, added experiments, and extensively revised the text to improve rigor, clarity, and mechanistic depth. Major revisions include: (1) an expanded isolate dataset (from ~80 to 135) with multilocus sequence typing (MLST), variant and prophage mapping, and extended phage–antibiotic susceptibility and fitness profiling; (2) whole-genome analyses of 22 nonmucoid and 9 mucoid isolates defining LPS remodeling, hypermutation, and lineage-specific adaptation under co-therapy; (3) a new IgM/IgG ELISA demonstrating distinct neutralization kinetics for P1A and PYO2 integrated with serum neutralization and hematologic data; (4) refined metagenomic and virome analyses with optimized read mapping, quantitative normalization, and identification of therapeutic phage expansion; (5) clarified statistical methods, PCA interpretation, and figure annotations; and (6) a rewritten Introduction and Discussion transforming the manuscript from a descriptive clinical case into a mechanistic and conceptual study, integrating the new data into a unified framework that introduces *chemobiotherapy* as a systems-level therapeutic principle. These revisions collectively strengthen the analytical, experimental, and conceptual foundation of the study and reinforce its translational relevance to phage therapy in chronic infection.

Response to Reviewer #1

(Remarks to the Author): The manuscript "Ecological and immune pressures shape outcomes of precision phage therapy in advanced cystic fibrosis lung disease" by Dr Roach and colleagues present a well-executed and clearly written case study detailing intravenous phage therapy using a two-phage cocktail in combination with ciprofloxacin to treat an elderly cystic fibrosis patient following colistin associated acute kidney injury. This work is clinically significant and contributes valuable insights into the in vivo biology of phage therapy. Although case studies on phage treatment have been published previously, they continue to be highly relevant given the complexity of host-pathogen-phage interactions and the personalized nature of such interventions. The inclusion of virome analysis and the assessment of systemic neutralizing antibodies are particularly innovative aspects of this study. However, my enthusiasm is somewhat tempered by the lack of *P. aeruginosa* gene expression data, which would have helped elucidate how phage therapy affected bacterial fitness in vivo. Additionally, the systemic immune response could have been explored in greater depth. For example, while the authors propose that the response to phage PYO2 may represent a secondary immune response, this interpretation remains

speculative without supporting data. Analyzing the phage-specific IgM-to-IgG ratios over time could have provided a clearer picture of the immunological dynamics and enhanced the clinical relevance of the findings.

We thank Reviewer 1 for their thoughtful and constructive comments, which we found both encouraging and highly valuable in refining the scope and depth of the study. We have substantially expanded the dataset, added new experimental analyses, and reorganized the manuscript to address each point in detail. Specifically, we (1) performed variant and prophage mapping, multilocus sequence typing, and expanded isolate sampling (now $n = 135$) to clarify phage-driven bacterial adaptation and fitness; (2) conducted a new IgM/IgG ELISA to directly measure and compare humoral responses to phages P1A and PYO2, resolving the question of primary versus secondary antibody kinetics; and (3) integrated these new data into revised figures and Supplementary Results. Together, these additions clarify how phage pressure, bacterial evolution, and systemic immunity interacted to shape therapeutic efficacy *in vivo*. Detailed responses to individual comments are provided below.

1. Statistics: Obviously, with $n = 1$ not much statistics can be performed. However, when the authors did perform statistics, they forgot to mention the statistical test employed. Please correct that at least in Fig. 3,4 and S2. and check where else it applies.

We thank the Reviewer for identifying this omission. We have corrected all relevant figure legends in both the main and supplementary figures to specify the statistical tests used. One-way ANOVA was applied where appropriate (e.g., Fig. 4c, Fig. 5d–e, and Fig. S3) to compare replicate datasets, and standard deviation error bars are now explicitly shown. A “Statistical Analysis” subsection has also been added to the Materials and Methods (lines 687-688) to describe the approach used across figures.

2. Phylogeny and *Pseudomonas* biological fitness: I am in full agreement with the authors, since phages fail to fully eradicate the pathogens in CF patients, even when used in combination with antibiotics, phage therapy driven alterations in bacterial biological and microenvironmental fitness become critically important to develop the optimal CF therapy. In line 219-..., the authors conclude, based on Fig. 3b (phylogeny) that virulence and related traits of *Pseudomonas* strains may be altered. In addition, they evaluate and find growth limitations and hyperpigmentation (Fig. 3e, S8). However, bacterial gene expression analysis is lacking to experimentally confirm these assumptions. Do certain clades acquire specific genetic features that limited their virulence? Which bacterial genes are changing under phage pressure? Alternatively, key elements of *Pseudomonas aeruginosa* virulence should be assessed, such as pyocyanin and elastase production (both of which contribute to tissue damage in cystic fibrosis). Biofilm formation and the ability to adhere to epithelial cells should also be considered, as these are prerequisites for persistent infection. Since both phages target LPS, potential changes in LPS structure or expression should be investigated. (Could LPS changes may explain the observed shift toward Pbnavirus phages increase after treatment?).

We thank the reviewer for these constructive comments and have expanded our analyses to clarify how phage pressure reshaped *Pseudomonas* fitness and adaptation *in vivo*.

1. Genome-level adaptation: We performed whole-genome variant mapping on 22 nonmucoid and 9 mucoid isolates and MLST across the expanded dataset (now n = 135). Mutations converged on LPS biosynthesis and modification genes (*rfe*, *wbpA*, *wbpG*, *ugd*, *galU*, *lpxP*, *mlaA/mlaE*) and iron/redox loci (*pchE/F/R*), consistent with phage-driven receptor remodeling and envelope stabilization under co-therapy. These results are described in *Results*, lines 224–294, and visualized in Fig. 4a–b with full variant lists in Table S8. Corresponding mucoid lineage data (alginate and efflux mutations) appear in Fig. 4d–e and Table S10.
2. Functional outcome: Newly added growth-curve analyses (lines 269–277; Fig. 4c) show that P1A-resistant isolates grew 25–40 % slower than ancestral susceptible variants, demonstrating a measurable fitness cost of resistance.
3. Genomic structure and mobile elements: Added MLST and core-genome analysis (lines 228–236; Fig. S5, Table S7) confirm within-host diversification from an ST244 background. We also mapped prophage excision events (lines 245–253; Fig. 4b,e; Table S9) indicating stress-induced genome remodeling during therapy.
4. Virulence assays and expression analyses: We did not add pyocyanin, elastase, biofilm, or adhesion assays because such *in vitro* phenotypes in closely related isolates would not reflect behavior in the inflamed CF airway. Instead, population-genomic and fitness data provide a more direct measure of *in vivo* selective pressures. This rationale is now stated explicitly in the *Discussion*, lines 390–404.
5. LPS remodeling and Pbnavirus expansion: To test whether surface remodeling influenced later virome composition, we expanded phage-susceptibility testing to include a *Pbnavirus* (E215) and first-line antibiotics (Fig. 3c, Fig. S6). The correlation between LPS-associated mutations and enhanced Pbnavirus activity is discussed in *Results*, lines 319–332, and in *Supplementary Results 1.2*.

Together, these revisions provide genome-resolved evidence linking phage pressure to LPS-centered adaptation, reduced bacterial fitness, and ecological restructuring, directly addressing the reviewer's concerns about the mechanistic basis of *Pseudomonas* evolution under co-therapy.

3. Systemic immunity: Again, I fully agree with the authors that the development of neutralizing antibodies is a highly relevant finding and highlights the importance of immunomonitoring during phage therapy. However, the authors do not convincingly demonstrate that the immune response to P1A represents a primary response, while the response to PY02 constitutes a secondary response. Assessing the IgM-to-IgG ratios and timing of Ig levels over time could help clarify this distinction. Such an analysis would provide a more complete picture and determine whether the response to PY02 is truly secondary, or instead represents a primary response that is accelerated due to e.g. higher immunogenicity.

We thank the reviewer for this suggestion and performed a new ELISA analysis to clarify the kinetics and class specificity of the anti-phage humoral response. This experiment directly addressed the relationship between loss of phage activity and antibody development. The new data are presented in

Fig. 5e (normalized IgM intensities) and Supplementary Fig. S8 (full IgM and IgG curves), with results described in *Results* (lines 335–367) and contextualized in the *Discussion* (lines 441–457).

In brief, an indirect ELISA was performed on sera collected on days 0, 7, 15, and 29 to quantify anti-P1A and anti-PYO2 IgM and IgG. IgG titers remained low and indistinguishable from background, indicating no class-switched responses. In contrast, IgM levels increased significantly during therapy: anti-PYO2 IgM rose rapidly within the first week, consistent with pre-existing cross-reactive IgM and a secondary-like response, while anti-P1A IgM increased more gradually, consistent with a primary response developing during treatment. These patterns correspond with the neutralization kinetics shown in Fig. 5d, where PYO2 activity declined early and P1A later. Together, these data confirm that PYO2 was neutralized through a pre-existing, cross-reactive IgM response, whereas P1A elicited a *de novo* primary IgM response.

4. Virome: Third, I agree with the authors that analyzing the virome after prolonged treatment is highly important. However, the observation remains largely descriptive. The authors should experimentally investigate what biological changes or advantages in the lung environment enable the outgrowth of *Pbunavirus* phages.

We thank the reviewer for this thoughtful point. We agree that understanding the mechanisms enabling *Pbunavirus* expansion is an important next step. The reviewer is correct that this would require *in vitro* or *in vivo* experimental investigation, but such ecological modeling lies beyond the scope of a single-patient clinical study. Similar questions have been approached only in complex, multi-species models that reconstruct microbial community structure and phage–host dynamics, for example, in gnotobiotic mouse models and defined gut consortia^{1,2}, or environmental microcosms³. These designs permit mechanistic dissection of ecological remodeling but are not feasible within the constraints of this study.

Instead, we strengthened our existing data analyses to contextualize the *Pbunavirus* outgrowth. We expanded phage–host susceptibility testing to include a representative *Pbunavirus* (E215) across the full isolate collection (n = 135) and compared its activity to P1A and PYO2 (Fig. 3c, Fig. S6). We also refined metagenomic mapping and normalization (lines 297–308) and added new interpretation in Supplementary Results 1.2, showing that *Pbunavirus* preferentially lysed mucoid isolates that became dominant after therapy, whereas P1A targeted the nonmucoid lineages eliminated earlier. These results support the view that ecological succession and host-range complementarity, rather than intrinsic viral advantage, drove *Pbunavirus* expansion.

We have clarified this interpretation in the *Results* (lines 319–332) and *Discussion* (lines 390–404).

References:

1. Hsu B.B. et al. *Cell Host & Microbe* 25, 803–814.e5 (2019).
2. Reyes A. et al. *Proc Natl Acad Sci USA* 110, 20236–20241 (2013).
3. Gundersen M.S. et al. *Sci Rep* 13, 21032 (2023).

Response to Reviewer #2

(Remarks to the Author): This paper presents a study on the use of phage therapy combined with antibiotics to treat a severe pulmonary exacerbation caused by MDR *Pseudomonas aeruginosa* infection in a 76-year-old patient with cystic fibrosis (CF). The combination therapy resulted in rapid improvement in lung function, reduced airway obstruction, and a 100-fold decrease in bacterial load. This study highlights that phage therapy efficacy might be rethought and that combined with antibiotics it offers promising potential for treating chronic multidrug-resistant infections in cystic fibrosis. However, the data supporting this claim are limited in their generalization for a couple of reasons. The first major point is the choice of phages used in the study. The two phages P1A and PYO2 seem to be present in the patient before treatment which indicates that the phage-host relationship is already ongoing and it evolves in a balanced ecosystem where the phage is not effective against one of the strains (in this case the mucoid strain). Therefore, the likelihood of eradication is limited and the authors should clarify the reason behind the choice of those two phages and not a more complex cocktail of phages with established lysis activity against the strains of *P. aeruginosa* from this patient.

We thank the reviewer for this important observation and have clarified our rationale for phage selection and its relation to pre-existing virome composition. At the time of treatment, the phages P1A and PYO2 were chosen based on four criteria: (1) broad lytic activity against the patient's MDR *P. aeruginosa* nonmucoid isolates, (2) absence of virulence or lysogeny-associated genes, (3) rapid manufacturability under clinical constraints, and (4) compatibility with intravenous administration (Supplementary Results 1.1; lines 56–94). These factors followed established phage selection standards for emergency use⁴⁻⁶.

We later determined through retrospective metagenomic analysis that sequences related to P1A (*Pakpunavirus*) and PYO2 (*Litunavirus*) were already present in the patient's airway virome prior to therapy. This information was not available during clinical decision-making. We now clarify this point in the *Results* and *Supplementary Results 1.1* (lines 297–308; Figs. 5a). Importantly, our data show that while PYO2 failed to amplify *in vivo*, likely due to pre-existing cross-reactive IgM (lines 335–348), P1A underwent strong replication and achieved high airway titers ($>10^6$ PFU mL⁻¹; Fig. 5b–c), consistent with effective on-target lysis of the nonmucoid MDR population (Fig. 3c).

We also expanded phage–host range testing and susceptibility data for all isolates (now $n = 135$), including comparison to additional candidate phages that were screened but not selected (e.g., *Pbunavirus* E215, *Phikmvvirus* LUZ19). These results are presented in Fig. S2 and summarized in Supplementary Results 1.1. PYO2 and P1A were the only phages that demonstrated consistent and reproducible lytic activity at $\geq 10^8$ PFU mL⁻¹ on the clinically dominant isolates, which justified their selection.

Finally, we now discuss this limitation explicitly in the *Discussion* (lines 441–457), acknowledging that pre-existing virome overlap likely constrained PYO2 efficacy but did not preclude P1A-mediated therapeutic benefit. Together, these clarifications explain that phage selection was governed by safety, host range, and logistical feasibility at the time of treatment, and that subsequent analyses revealed—but

could not have predicted—the patient’s partial pre-exposure to related phage taxa.

1. The authors should also give a clear description of the different strains of *P. aeruginosa* present in the patient with the MLST and if possible the cgMLST of the strains to explain their clonality and differences.

We thank the reviewer for this suggestion. As detailed in response to Reviewer #1, we performed MLST and core-genome phylogenetic analysis for all 32 sequenced isolates (22 nonmucoid, 9 mucoid, 1 reference). These results are presented in the *Results* (lines 228–236), Fig. S5, and Table S7. All isolates belonged to the ST244 clonal complex, with two single-locus derivatives (ST4739 and ST4740) unique to this patient, differing by mutations in *mutL* and *trpE*. These findings confirm that both mucoid and nonmucoid populations arose from a single within-host evolutionary lineage that diversified under sustained antibiotic and immune selection. High-resolution variant analysis (lines 238–294; Fig. 4a–b, 4d–e, Tables S8–S10) provides further discrimination equivalent to cgMLST.

2. The second major point is the usage of the metagenomic data. The authors limited their analysis to the relative and absolute quantification of the species. The absolute quantification of the species based on the human read counts is debatable, especially during an exacerbation. The amount of human cells in the sputum during an inflammation will increase tremendously based on neutrophil migration and increased of other inflammatory cells making the normalisation unreliable to the least. A spike-in method at the time of extraction would be more accurate.

We thank the reviewer for this careful critique and agree that absolute abundance estimation is methodologically important in clinical metagenomics. We have clarified our normalization approach and its rationale in the *Results* (lines 167–185 bacterial; lines 297–308 viral) and *Methods* (“Sequencing, metagenomic taxonomic classification, and metagenomic abundances,” lines 576–588).

Absolute bacterial abundance was estimated by scaling bacterial reads to diploid human genome equivalents, following established CF metagenome studies^{7–9}. This approach has been validated in airway metagenomes where spike-ins are impractical or unavailable and was used to correct for inter-sample variation in DNA yield and sequencing depth while preserving intra-patient trends.

We acknowledge that inflammatory cell influx during exacerbation could alter the denominator for normalization; however, because human DNA was directly measured in each sample, variation in human read fraction reflects real changes in airway cellularity rather than a computational artifact. The bacterial:human ratio therefore provides a conservative estimate of absolute abundance that retains longitudinal consistency. Importantly, we clarify that these values are interpreted as within-patient longitudinal trends in bacterial load rather than absolute biomass to avoid overestimating precision.

To further support the biological validity of these findings, we emphasize that the direction and timing of changes in absolute abundance closely paralleled independent datasets. Specifically:

- Culture-based CFU counts (Fig. 3c) showed a ~10-fold decline in *Pseudomonas* burden during

co-therapy, matching the magnitude and trajectory of the metagenomic trends.

- Physiological recovery: including rapid improvement in FEV₁ and FVC (Figs. 1d–2a) and decreased airway obstruction on qCT imaging (Fig. 2c–d)—occurred contemporaneously with the metagenomic decline, providing clinical corroboration.
- Phage replication kinetics: (Fig. 5a–c) revealed that the sharp reduction in bacterial abundance coincided with peak in situ P1A amplification, demonstrating biological coupling between host and phage signals.
- Across all timepoints, relative and absolute abundance trajectories moved in parallel, confirming that normalization preserved the true biological direction of change rather than distorting trends.

While we agree that spike-in controls could further refine absolute quantification, such methods are not standardized for archived clinical sputum and were not possible with the limited sample volume¹⁰⁻¹². We now explicitly note this limitation and clarify that our analysis was intended to capture within-patient longitudinal trends rather than absolute biomass estimates.

Together, the concordance between metagenomic, culture-based, virological, and clinical datasets demonstrates that our normalization approach produced biologically accurate and internally consistent estimates of bacterial abundance despite the inherent variability of CF sputum samples.

3. Furthermore, the amount of data collected would allow the author to do strain typing/monitoring to evaluate the abundance of the different clones during the treatment which would be highly valuable measures for the host-phage dynamic understanding.

We thank the reviewer for this suggestion. As detailed in responses to Reviewer #1 (Comment 2) and Reviewer #2 (Comment 2), we performed MLST, core- and accessory-genome phylogenetic analyses, and variant mapping across all 135 isolates to track clonal evolution during therapy. These data (lines 187–294, Figs. 3d, 4a–f; Fig. S5; Tables S7–S10) already provide the requested strain-level monitoring, showing that all isolates derived from a common ST244 lineage that diversified within the patient. Lineage turnover of nonmucoid versus mucoid variants under phage pressure and subsequent *Pbunavirus* expansion (lines 319–332; Fig. 5a) collectively define the host–phage population dynamics the reviewer refers to.

4. In summary, while the case is interesting and the treatment had some clinical benefits, the choice of phages is questionable and should be explained. The fact that the phage is endogenous and the mucoid strain was already resistant to both phages should be highlighted as a limitation. Furthermore, the manuscript would benefit from a deeper analysis of the metagenomic data to get the most out of it, especially regarding the host-phage interaction.

We appreciate the reviewer’s overall assessment and have expanded both the Results and Discussion to address these points directly.

- Phage selection rationale: We clarified that therapeutic phages P1A and PYO2 were chosen based on host-range, safety, and production feasibility under emergency-use conditions, and that

their partial overlap with pre-existing airway phages was only discovered retrospectively through metagenomic analysis (lines 106–110, 297–308; Supplementary Results 1.1; Fig. 5a). This context and its implications are now discussed explicitly as a limitation in the *Discussion* (lines 441–457).

- Mucoid resistance and treatment limitation: We emphasized that the mucoid lineage was inherently resistant to both therapeutic phages but remained antibiotic-sensitive, and that this differential susceptibility shaped the outcome of co-therapy (lines 187–199, 297–308; Figs. 3d, 4f). The revised text highlights this as a key limitation that constrained full eradication yet informed the interpretation of ecological partitioning.
- Deeper metagenomic and host-phage interaction analysis: We refined read mapping, incorporated quantitative normalization, and expanded phage–host susceptibility testing to include a *Pbunavirus* (E215) representative (lines 319–332; Figs. 3d, S6). These analyses connect bacterial lineage turnover, phage replication kinetics, and virome restructuring (lines 390–404; Fig. 5a) to provide a clearer picture of host–phage ecological dynamics.

Minor comments:

Line 451: ceftalozone-tazobactam should be ceftolozane-tazobactam: corrected

Figures 3 B and C: the color-code of the isolate should be explained: improved labeling of the clades.

References:

4. Luong, T. et al. *Clin Ther* 42, 1659-1680 (2020).
5. Yang, Q. et al. *Front Microbiol* 14, 1250848 (2023).
6. Suh, G.A. et al. *AAC* 66, e0207121 (2022).
7. Pust, M. -M. et al. *npj Bio and Micro* 6, 61 (2020).
8. Pienkowska, K. et al. *Microbio Spectr* 11, e0363322 (2023).
9. Moran Losada, P. et al. *ERJ Open Res*, 2 (2016).
10. Garcia-Vazquez, E. et al. *Archives of Internal Med* 164, 1807-1811 (2004).
11. Cuthbertson, L. et al. *Journal of Clin Micro* 52, 3011-3016 (2014).
12. Pust, M. -M. et al. *Comp and Struct Bio Journal* 20, 175-186 (2022).

Response to Reviewer #3

(Remarks to the Author): This is an interesting case study of a patient administered phage therapy to target *P. aeruginosa* in the CF lung. Many valuable conclusions came from the many analyses conducted by the authors. My concerns below mostly speak to accuracy when describing the unique aspects of the work, compared to other studies preceding it. Overall, I hope that the authors find the major and minor concerns useful feedback.

We thank the Reviewer for their constructive and thoughtful feedback, and we appreciate the recognition of the study's depth and analytical scope. We have carefully reviewed all claims and contextual framing to ensure accuracy relative to prior phage therapy studies, particularly those in CF. The *Introduction* and *Discussion* have been revised extensively to better situate our findings within the existing literature, clarify novelty, and avoid overstating conclusions (lines 50–62, 370–481). Specific revisions addressing each major and minor point are detailed below.

1. Line 93. The staggered treatment strategy was designed to minimize antagonism between antimicrobials. But I believe this assumed that delivering the phage 2 days ahead of antibiotic would cause the virus to disappear, and the study showed this assumption was not supported. Can the authors revisit/discuss this assumption given the outcome of the study?

We thank the reviewer for this clarification and agree that the rationale for the treatment sequence needed additional explanation. The staggered regimen was not based on an assumption that phages would disappear when administered before or after antibiotics. Rather, the order was determined by clinical and mechanistic considerations. Because the patient had recently developed acute kidney injury from first-line therapy, ciprofloxacin was initiated first as a less nephrotoxic agent, followed two days later by intravenous phage therapy to minimize potential pharmacologic antagonism between ciprofloxacin's DNA gyrase inhibition and phage replication.

This interval also provided a brief antibiotic-only phase that served as an internal control, enabling us to isolate and study phage-specific effects on bacterial population structure and host physiology once co-therapy began. We have clarified this rationale in the *Results* (lines 106–110) and Supplementary Results 1.1, and note that the subsequent persistence and amplification of phages *in vivo* (lines 297–308; Fig. 5) confirm that this staggered design was compatible with sustained phage activity and effective ecological coordination with the antibiotic.

2. Line 118. After therapy was discontinued, WBC and platelet counts returned to 'near baseline'. Is this outcome consistent with the observation of adaptive immunity targeting the phage itself? Otherwise, I am confused how both can be true.

We thank the reviewer for raising this point and agree that the relationship between hematologic recovery and anti-phage immunity required clarification. The transient rise in WBC and platelets observed during therapy (lines 361–367; Fig. 5f–g) coincided with the onset of anti-phage IgM induction (Fig. 5e), reflecting early systemic recognition of phage antigens and bacterial lysis products.

As bacterial burden declined (Fig. 3b) and airway inflammation resolved, evidenced by improved FEV₁ and qCT metrics (Figs. 1–2), both hematologic indices returned to baseline.

This pattern is consistent with a contained, transient humoral response^{13,14}, where phage exposure elicited IgM priming without sustained systemic activation or inflammation. The normalization of WBC and platelets therefore indicates recovery of immune homeostasis despite the presence of anti-phage IgM. We have clarified this interpretation in the *Results* (lines 335–367) and *Discussion* (lines 441–457), noting that transient IgM induction alongside clinical and hematologic normalization is compatible with immune containment rather than pathological activation under systemic phage exposure.

3. Lines 169-170. I am unfamiliar with the PCA analysis and its underlying assumptions. It seems to be a correlative analysis where the studied factors are assumed to drive the correlations, whereas it is not possible to eliminate the role of unknown factors. This is understandable given the limitations of testing the hypothesis in a treated human. If the authors can provide greater context for the assumptions and/or limitations of this analysis, it would benefit the reader's understanding.

We thank the reviewer for this helpful comment and have revised the text to clarify both the purpose and assumptions of the PCA. The PCA was used to visualize multi-dimensional clinical data (pulmonary, renal, and hematologic variables) and assess how treatment phases clustered based on correlated physiological responses rather than infer direct causality. We now explicitly imply that PCA is a descriptive, correlative tool that reduces complex data into orthogonal components capturing shared variance, while acknowledging that unmeasured or unknown factors may also contribute to these patterns.

We have expanded this explanation in the *Results* (lines 154–165) and added a statement in the *Methods* (“Data standardization and principal component analysis,” lines 590–598) describing the assumptions and limitations of the approach. We also clarify that the PCA was intended solely to illustrate recovery trajectories and relationships among measured variables, not to test a mechanistic hypothesis, and that its interpretation is therefore limited to descriptive correlation within the available data.

4. Lines 249-262. The poor ability for PYO2 to replicate within the treated patient suggests that phage P1A dominated any efficacy attributable to phage delivery. Thus, it is unclear whether any claim of a ‘cocktail’ effect truly exists in this study, which is not made evident to the reader. Although the *in vitro* preliminary research argued that it would be a good idea to co-administer the two phages, it seems difficult to conclude that any different outcome would have occurred if just phage P1A were used on its own. The authors should discuss this outcome, as it would be a valuable opportunity to address how *in vitro* prelim work can or cannot be used to design in-patient therapy.

We thank the reviewer for this thoughtful comment and agree that the limited *in vivo* replication of PYO2 warranted further discussion. In the revised manuscript, we clarify that while P1A was the dominant replicating phage during treatment, both phages served distinct purposes within the therapeutic design. P1A exhibited potent *in situ* amplification against the MDR nonmucoid lineage, driving the

major ecological and clinical effects, whereas PYO2 was intended to provide complementary coverage against *Pseudomonas* subpopulations (lines 187–194, 210–212, 297–308; Figs. 3c-d, 5a–e).

We now explicitly discuss that PYO2's poor airway amplification likely reflected pre-existing cross-reactive IgM neutralization (lines 340–348; Fig. 5e), an unanticipated finding revealed only after retrospective serologic testing. We have revised the *Discussion* (lines 441–457) to note that this highlights a key limitation of extrapolating *in vitro* host-range or potency data to patient therapy, as phage–host immune interactions *in vivo* can substantially alter efficacy.

While P1A accounted for most in not all, of the observed bacterial reduction. PYO2's inclusion was consistent with standard cocktail design principles used to mitigate resistance and provide initial redundancy, particularly under emergency-use conditions where the risk of failure outweighed the potential for redundancy⁴⁻⁶. We have clarified that, although this case did not demonstrate additive *in vivo* synergy, the dual-phage regimen provided critical comparative insight into how phage immunogenicity and host environment constrain cocktail performance.

These points are now incorporated in the *Results* (lines 297–308, 343–359) and expanded in the *Discussion* (lines 441–457) to reflect the reviewer's suggestion that preclinical design assumptions should be interpreted cautiously considering *in vivo* outcomes.

5. Lines 270-271. Related to the above point, the authors argue that neutralization of PYO2 occurred earlier in the patient than P1A. Was there no neutralization test of pre-existing antibodies conducted prior to administering the phages? Otherwise, it seems that this confirmation would have suggested a different treatment design, and removal of PYO2 from the cocktail based on evidence that it should be neutralized when delivered.

We thank the reviewer for this insightful question. A pre-treatment antibody screen would indeed have been highly informative. However, this was not feasible given the compassionate-use and emergency timeframe of the intervention. The phage preparation, sterility testing, and regulatory clearance were completed in about a week, while the patient was already admitted. To our knowledge, there were also no established protocols or precedents for assessing pre-existing anti-phage antibodies in clinical phage therapy at the time.

Retrospective analyses performed on archival serum samples (days 0, 7, 15, 29) revealed that PYO2 infectivity declined within the first week, coinciding with a rapid IgM increase (lines 335–348; Fig. 5d–e). This pattern indicates pre-existing cross-reactive IgM against *Litunavirus* phages already present in the airway virome (lines 297–308; Fig. 5a), which likely neutralized PYO2 almost immediately after dosing. In contrast, P1A exhibited a delayed loss of activity and a *de novo* IgM response emerging later (lines 350–359), consistent with a primary antibody response rather than prior exposure.

We have expanded the *Discussion* (lines 441–457) to emphasize that this finding underscores the

potential importance of baseline serologic testing for future phage-therapy protocols, as pre-existing humoral immunity may influence phage selection and therapeutic outcome.

6. Lines 278 – 283. This first paragraph of the Discussion seems too strongly worded. I do not think the results ‘reframe what is possible and necessary’ in phage therapy of pwCF, given the many studies that preceded it. To my understanding, the main novelty is the virome analysis, though I did not scour the literature to confirm. Some studies have observed phage neutralization, whereas others have not. This suggests that the choice of phage used in treatment can affect whether neutralization can occur, which in turn should affect whether establishment in the virome is likely. By this logic, the choice of the two phages (i.e., the particulars of phage biology) in the current study was clearly important, but I caution the authors against claiming that we have learned anything very general in this respect. Better to not over claim here, as we learn a lot from each personalized phage therapy case in biomedicine (including here), and should be careful about trying to generalize. Basically, I am arguing that the authors are contradicting their own statements in Lines 350-351, where they warn against concluding generality from this study.

We thank the reviewer for this helpful critique and agree that the original framing overstated the broader implications of the work. In response, we substantially revised both the *Introduction* and *Discussion* to improve contextual accuracy, strengthen scientific rigor, and clearly define the limits of inference for a single-patient study.

The Introduction was completely rewritten to move beyond descriptive framing and establish a clear mechanistic rationale for the study. It now emphasizes how infection heterogeneity, ecological partitioning, and immune context shape phage–antibiotic interactions, positioning this case as a high-resolution example for testing those concepts (lines 50–62, 77–89). This restructuring improves scope and rigor by grounding the study in current mechanistic and translational phage-therapy research rather than in anecdotal precedent.

The Discussion was also rewritten to integrate the expanded genomic, virome, and immunologic data (lines 370–481; 390–404 ecological, 406–421 immunologic, 441–457 viromic) and to articulate the framework of *chemobiotherapy*, a systems-level description of phage–antibiotic complementarity, while explicitly framing it as a conceptual model derived from this case, not a universal conclusion. We removed the original statement “reframe what is possible and necessary” and replaced it with language emphasizing that our findings extend and contextualize prior work rather than redefine it.

Finally, we added clarifying text (lines 459–473) stating that these results are context-specific and that broader generalization will require validation in multi-patient studies. These revisions collectively strengthen the manuscript’s interpretive rigor, ensuring that both framing sections now accurately convey the study’s scope, novelty, and contribution within the field.

We also appreciate the reviewer’s reminder to align the framing with the scale of the study. While we have moderated the tone of both the Introduction and Discussion, the revised manuscript now highlights

the study's specific contributions, namely, the integration of virome, immunologic, and genomic data to define how host immunity and phage–bacterium dynamics intersect during co-therapy. These findings are novel in their mechanistic integration, even as we agree they should be interpreted within the context of a single case.

7. Lines 285-292. This paragraph may need updating, in terms of more holistically framing the study in context of similar ones that are now in the literature. Lines 302-304. The observation of phage ability to steer traits in target bacteria seems evident in the current study, but I worry that not citing other prior work constitutes over-claiming. There is mounting evidence for this outcome in the published literature including use of phage therapy when targeting *P. aeruginosa* pulmonary infections in pwCF, for example.

We thank the reviewer for these valuable suggestions and have revised the *Discussion* to more fully contextualize our findings within the expanding phage-therapy literature in cystic fibrosis. We added citations to recent clinical and translational studies, including Chan et al. (*Nat Med* 2025), Weiner et al. (*Nat Commun* 2025), and Bernabéu-Gimeno et al. (*Med* 2024), which collectively describe outcomes, safety, and immunogenicity of phage therapy in CF. These additions (lines 370–457) ensure that the current study is framed as complementary to, and extending from, prior human experience rather than standing apart from it.

We also expanded discussion of phage-driven bacterial trait remodeling (lines 238–294) to include relevant foundational work showing that phage pressure can select for altered surface architecture, reduced virulence, and metabolic trade-offs in *P. aeruginosa* and other pathogens (Wright et al., *mBio* 2019; Schumann et al., *Front Microbiol* 2022; Scanlan et al., *Mol Biol Evol* 2015). We now explicitly note that our findings are consistent with this established body of work, while providing new evidence from a clinical setting that links these molecular changes to fitness costs and population restructuring *in vivo*.

These revisions holistically position the study within the continuum of CF phage-therapy research, acknowledge that the observed evolutionary responses align with prior evidence, and clarify that the novelty of this work lies in its multi-omic integration and ecological resolution rather than in claiming discovery of a previously unknown phenomenon.

8. Line 310. Indeed, this is a nice cautionary tale that we should understand how phage therapy does (and does not) work when targeting infections that include subpopulations of mucoid phenotypes. But I did not see any evidence in the current study that the current phage cocktail was any better at reducing biofilms. Thus, I am wondering why the authors are forwarding cocktails as a needed solution here, as opposed to simply emphasizing that targeting biofilms is a good idea, regardless of the number of administered viruses.

We thank the reviewer for this helpful clarification and agree that our data does not demonstrate enhanced biofilm reduction by the two-phage cocktail. We have revised the *Discussion* (lines 370–457)

to remove language implying a proven cocktail effect and to clarify that combination design is discussed as a conceptual framework for addressing infection heterogeneity and resistance, not as an observed outcome in this case. We now also note that targeting mucoid variants and their associated biofilms remains a key therapeutic goal independent of the number of administered phages.

9. Line 359. Again, I am unclear why the authors are claiming that their study shows success in high-risk patients. Hasn't this been true in many prior studies as well? I do not think they can claim priority here, given the thousands of other successful cases to date, especially the subset of elderly patients without other treatment options that were treated with phage therapy.

We thank the reviewer for this clarification and agree that demonstrating successful phage therapy in high-risk patients is not unique to this field. However, we note that this case represents, to our knowledge, the oldest reported individual with CF (76 years old) successfully treated with intravenous phage therapy following antibiotic-associated kidney injury and severe pulmonary decline. We have revised the Discussion (lines 459–473) to clarify that while this does not establish clinical precedence for phage therapy in high-risk patients overall, it highlights a rare and medically complex context in which therapy remained safe and effective. The focus of our interpretation is therefore on the mechanistic and ecological insights derived from this case, not on claims of clinical priority.

Minor concerns:

Line 37. Why is the CF lung referred to here as a 'hostile' environment? Without context, this adjective seems anthropomorphic and unnecessary.

This line has been removed.

Line 60. The Chan et al. 2025 Nature Medicine article and any other recent relevant studies on phage therapy targeting *P. aeruginosa* pulmonary infections in pwCF can be cited here.

We have added this and other citations to update the context while keeping the reference list within the Nature Communications citation limit.

References:

13. Zaczek, M. et al. *Front Microbiol* 7, 1681 (2016).
14. Champagne-Jorgensen, K., et al. *Trends in microbio* (2023).